# The Role of Soluble Low-Density Lipoprotein Receptor-Related Protein-1 in Obstructive Sleep Apnoea

**DOI:** 10.3390/jcm10071494

**Published:** 2021-04-03

**Authors:** Martina Meszaros, Laszlo Kunos, Adam Domonkos Tarnoki, David Laszlo Tarnoki, Zsofia Lazar, Andras Bikov

**Affiliations:** 1Department of Pulmonology, Semmelweis University, 1083 Budapest, Hungary; martina.meszaros1015@gmail.com (M.M.); laszlokunos@gmail.com (L.K.); zsofia.lazar@yahoo.com (Z.L.); 2Medical Imaging Centre, Semmelweis University, 1082 Budapest, Hungary; tarnoki2@gmail.com (A.D.T.); tarnoki4@gmail.com (D.L.T.); 3North West Lung Centre, Manchester University NHS Foundation Trust, Manchester M13 9WL, UK; 4Division of Infection, Immunity and Respiratory Medicine, University of Manchester, Manchester M13 9NT, UK

**Keywords:** sleep apnoea, OSA, intermittent hypoxia, inflammation

## Abstract

Intermittent hypoxia in obstructive sleep apnoea (OSA) is related to inflammation and metabolic abnormalities. Soluble low-density lipoprotein receptor-related protein-1 (sLRP-1) is involved in anti-inflammatory and metabolic processes. However, its ligand, calreticulin (CALR) promotes pro-inflammatory responses and apoptosis. Our aim was to analyse the levels of these biomarkers in OSA. We recruited 46 patients with OSA and 30 control subjects. Inpatient sleep study was performed and fasting plasma samples were collected. Triglyceride glucose index (TyG) and atherogenic index of plasma (AIP) were calculated. Plasma sLRP-1 levels were significantly lower in the OSA group compared to the controls (1.67 (0.90–2.11) mg/L vs. 1.99 (1.53–3.51) mg/L; *p* = 0.04) after adjustment for age, gender, BMI and lipid profile. Plasma sLRP-1 concentrations were inversely related to age (r = −0.29), BMI (r = −0.35), cigarette pack years (r = −0.31), LDL-C (r = −0.34) and triglyceride levels (r = −0.27), TyG (r = −0.37) and AIP (r = −0.27) as well as to the oxygen desaturation index (ODI, r = −0.24; all *p* < 0.05). BMI (*p* = 0.01) and ODI (*p* = 0.04) were independent predictors for low sLRP-1 levels. CALR did not differ significantly between the two groups (0.23 (0.17–0.34) ng/mL vs. 0.24 (0.20–0.36) ng/mL *p* = 0.76). We detected lower sLRP-1 levels in subjects with OSA which could contribute to metabolic abnormalities associated with this disease.

## 1. Introduction

Obstructive sleep apnoea (OSA) is the most prevalent sleep-related breathing disorder, which is characterised by a repetitive collapse of the upper airways during sleep and is associated with sleep fragmentation, intermittent hypoxia (IH), increased sympathetic tone, oxidative stress and systemic inflammation. These mechanisms play an important role in the pathogenesis of OSA and its comorbidities. OSA has emerged as an important risk factor for metabolic abnormalities, such as insulin resistance, dyslipidaemia and consequential atherosclerosis with a high prevalence of cardiovascular morbidity and mortality [1,2].

OSA causes metabolic abnormalities through several mechanisms. IH induces the expression of sterol regulatory element-binding protein-1 (SREBP-1), which regulates lipid biosynthesis resulting in an increased triglyceride secretion from the liver [3]. IH and sleep fragmentation elevates sympathetic outflow which stimulates lipolysis [4] and the release of free-fatty acids to the circulation [5]. Moreover, IH blocks the activity of peripheral lipoprotein lipase resulting in an impaired lipoprotein clearance [6]. The increased sympathetic tone also leads to decreased sensitivity to insulin and pancreatic β-cell dysfunction contributing to insulin resistance [2]. These abnormalities are further worsened by systemic inflammation, aberrant immune response and altered coagulation characterising OSA [2]. However, the exact mechanisms of OSA-related metabolic dysregulation have not been thoroughly explored [7].

Low-density lipoprotein receptor-related protein-1 (LRP-1, also known as CD91) is a transmembrane receptor of the low-density lipoprotein receptor (LDLR) family, which consists of an extracellular α-subunit (515-kDa) and a partly transmembrane β-subunit (85-kDa). LRP-1 is abundantly expressed in several tissues, such as the liver, lung or blood vessels with a multifactorial role [8]. The expression of LRP-1 is strongly influenced by inflammatory and metabolic processes [9]. Most importantly, SREBP-1 that is upregulated in OSA [3] decreases the expression of LRP-1 [10]. LRP-1 acts as an endocytic receptor of a wide range of ligands, such as α2-macroglobulin, mediating their internalisation [11]. The primary role of LRP-1 is the apolipoprotein E (ApoE)-mediated clearance of triglyceride-rich remnant lipoproteins in the liver [12]. Moreover, LRP-1 is also involved in other inflammatory and metabolic pathways [11] as well as the coagulation system [13] and exerts both pro-inflammatory and anti-inflammatory as well as anti-atherogenic properties [14]. Its soluble form (soluble LRP-1 = sLRP-1) is shed from LRP-1 and contains the α-subunit and a 55 kDa-fragment of the β-subunit of LRP-1. Notably, sLRP-1 can be measured in plasma samples [15].

Calreticulin (CALR) is an endoplasmic reticulum-associated chaperone which is exposed to the cell surface and even released extracellularly upon various stress stimuli, such as inflammation and hypoxia [16]. On the cellular surface it binds with LRP-1 of the immune cells and participates in the immunogenic cell death. The binding of CALR to LRP-1 leads to the induction of phagocytosis of apoptotic cells and this interaction promotes pro-inflammatory responses in macrophages [17]. OSA is characterised by a pro-apoptotic state [18,19,20], therefore blood calreticulin levels could theoretically be elevated in OSA.

Neither LRP-1 or CALR have been investigated in OSA before, despite that intermittent hypoxaemia, altered metabolism and systemic inflammation may change their levels, and these molecules have also the potential to affect these processes. Therefore, the main aim of the study was to investigate circulating sLRP-1 and CALR levels in OSA and relate them to markers of disease severity.

## 2. Materials and Methods

### 2.1. Study Design and Subjects

Seventy-six volunteers participated in this study. They were originally referred for a diagnostic sleep study due to suspected obstructive sleep apnoea (i.e., snoring, pauses in the breathing, daytime sleepiness). None of them had been previously diagnosed with OSA or had received any treatment for OSA, such as continuous positive airway pressure (CPAP), mandibular advancement device or upper airway surgery. Subjects who were older than 18 years and those who spent at least 4 h in sleep were eligible. Patients with malignancy within 10 years, infection within 2 months, autoimmune disorders, uncontrolled chronic disease (such as acute heart disease and respiratory failure or uncontrolled diabetes) or on statin therapy were excluded.

Following an inpatient diagnostic sleep study, systolic (SBP) and diastolic blood pressure (DBP) were measured, and venous blood was taken in the morning to evaluate plasma sLRP-1 and CALR, as well as serum glucose, total cholesterol, high-density lipoprotein cholesterol (HDL-C), low-density lipoprotein cholesterol (LDL-C), triglyceride, lipoprotein(a), apolipoprotein A1 (ApoA1), apolipoprotein B (ApoB) and C-reactive protein (CRP) levels under fasting conditions and before taking medications. Triglyceride glucose index (TyG), a marker of insulin resistance, was calculated as ln(TG(mg/dl)*glucose(mg/dl)/2). Atherogenic index of plasma (AIP), a biomarker of cardiovascular diseases, was calculated as log (TG(mmol/L)/HDL-C(mmol/L)). The participants filled out the Epworth Sleepiness Scale (ESS) and a detailed medical history was taken which included smoking habits and comorbidities. Comorbidities were defined according to the patients’ report, available medical history and current medications.

All procedures performed in studies involving human participants were in accordance with the ethical standards of the institutional and national research committee and with the 1964 Helsinki declaration and its later amendments or comparable ethical standards. The study protocol has been approved by the Semmelweis University Scientific Research Ethics Committee (TUKEB 30/2014, RKEB 172/2018) and all research was performed in accordance with relevant regulations. Written informed consent was provided by each volunteer.

### 2.2. Sleep Studies

Full-night inpatient cardiorespiratory polygraphy (PG, *n* = 37) and polysomnography (PSG, *n* = 39) were performed using the Somnoscreen Plus Tele RC and Somnoscreen PSG devices (Somnomedics GmbH Germany, Randersacker, Germany). Sleep stages, movements and cardiopulmonary events were scored manually according to the American Academy of Sleep Medicine (AASM) guidelines [21]. Apnoea was defined as at least a 90% decrease in airflow lasting for at least 10 s. Hypopnoea was defined as at least a 30% reduction in the nasal airflow lasting for more than 10 s with a ≥ 3% oxygen desaturation or an arousal. Total sleep time (TST), sleep period time (SPT) and minimal oxygen saturation (MinSatO_2_) were recorded, while apnoea-hypopnoea index (AHI), oxygen desaturation index (ODI) and the percentage of total sleep time with saturation below 90% (TST90%) were calculated to evaluate the severity of OSA. OSA was defined according to the International Classification of Sleep Disorders (Third Edition) criteria (i.e., AHI ≥ 5/h with daytime or nighttime symptoms or with the presence of comorbidities, or AHI ≥ 15/h irrespective of symptoms or comorbidities). Subjects with AHI < 5/h comprised the control group.

### 2.3. Biomarker Measurements

EDTA-treated venous blood samples were processed within 30 min after collection and centrifuged at 1500 rpm for 10 min at 4 °C. After centrifugation, plasma samples were separated and stored immediately at −80 °C until further analysis. Plasma sLRP-1 and CALR levels were measured using commercially available ELISA kits (Human Low Density Lipoprotein Receptor Related Protein 1 ELISA Kit from Bioassay Technology Laboratory, Shanghai Korain Biotech Co. Ltd. Inc, Shanghai, China (Catalogue number: E2298Hu)); Human Calreticulin (CRT) ELISA Kit from Cusabio Technology Llc., Houston, TX, USA (Catalogue number: CSB-E09787h)). Biomarker measurements were performed in duplicates according to the manufacturers’ instructions and the mean concentrations of the two measurements were recorded for analysis. The limit of detection was 0.027 mg/L for sLRP-1 and 0.039 ng/mL for calreticulin. All concentrations were above the detection limit. The intra-assay coefficients of variation were 9.27 ± 9.92% and 14.35 ± 12.17% for sLRP-1 and calreticulin, respectively.

### 2.4. Statistical Analysis

Statistical analyses were performed with JASP 0.11.1 (University of Amsterdam, Amsterdam, Netherlands) and Graph Pad Prism 5.0 (GraphPad Software, San Diego, CA, USA). The normality of the data was assessed with the Shapiro-Wilk test, which showed non-parametric distribution for sLRP-1 and calreticulin levels (both *p* < 0.01). Clinical data, categorical variables and biomarkers were compared between OSA and control groups with t-test, Mann-Whitney U-test and Chi-square test. We applied non-parametric ANCOVA after adjustments on age, gender, BMI and the lipid profile (total cholesterol, HDL-cholesterol, LDL-cholesterol, triglyceride, lipoprotein (a)) to evaluate the differences in sLRP-1 and calreticulin levels between OSA and control groups as well as among the severity groups (control patients, patients with mild, moderate and severe OSA). Plasma sLRP-1 and calreticulin levels were compared with clinical and sleep parameters with the non-parametric Spearman test. To further assess a potential relationship between biomarker levels and clinical variables, multivariate logistic regression analysis was performed, where sLRP-1 and CALR levels were grouped into low- and high sLRP-1 and CALR groups as dependent variable (according to the median 1.84 mg/L for sLRP-1 and 0.23 ng/mL for CALR). Data with parametric distribution were presented as mean ± standard deviation (SD). Data with non-parametric distribution were expressed as median with interquartile range (25–75 percentile). A *p*-value < 0.05 was considered significant.

The sample size was calculated to find a difference in sLRP-1 and CALR levels between the OSA and control groups in the adjusted model with an effect size of 0.50, power of 0.80 and a probability of α error of 0.05 [22].

## 3. Results

### 3.1. Patient Characteristics

Forty-six patients were diagnosed with OSA. Among them, 13 had mild (AHI 5–14.9/h), 14 had moderate (AHI 15–29.9/h) and 19 had severe (AHI ≥ 30/h) disease. Patient characteristics and comparisons between the two groups are shown in Table 1. Patients with OSA were significant older and had higher prevalence of males, hypertension and smokers (all *p* < 0.01). In the OSA group significantly higher SBP and DBP values, higher CRP, glucose and triglyceride levels, higher AIP and TyG values and lower HDL-C levels were measured compared to the control group (all *p* < 0.01). Patients with OSA had higher AHI, ODI and TST90% and lower minSatO_2_ (all *p* < 0.01). Patient characteristics and comparisons between the severity groups among patients with OSA are shown in Appendix A (Please see the Appendix A).

### 3.2. Plasma sLRP-1 Levels between Control and OSA Groups

Plasma sLRP-1 levels were significantly lower in the OSA group compared to the controls (1.67 (0.90–2.11) mg/L vs. 1.99 (1.53–3.51) mg/L; *p* = 0.04; Figure 1). However, the difference between the severity groups was not significant (*p* = 0.15, Figure 2).

### 3.3. Association between Circulating sLRP-1 Levels and Clinical Variables

Circulating sLRP-1 levels inversely related to age (r = −0.29, *p* = 0.01), BMI (r = −0.35, *p* < 0.01), cigarette pack years (r = −0.31, *p* < 0.01), LDL-cholesterol (r = −0.23, *p* = 0.04), triglyceride (r = −0.27, *p* = 0.02), TyG (r = −0.37, *p* <0.01) and AIP (r = −0.27, *p* = 0.02). There was a significant inverse relationship between sLRP-1 concentrations and ODI (r = −0.23, *p* = 0.04), however there was no correlation between sLRP-1 and the other markers of sleep architecture (all *p* > 0.05). The results of the Spearman correlation are shown in Table 2. Using multivariate regression analysis, we found that BMI (β = −0.16, *p* = 0.01) and ODI (β = −0.04, *p* = 0.04) were independently associated with lower sLRP-1 levels. There was no correlation between sLRP-1 and CALR levels (r = −0.17, *p* = 0.15).

### 3.4. Circulating Calreticulin Levels

Plasma CALR levels did not differ significantly between the OSA and control groups (0.23 (0.17–0.34) ng/mL vs. 0.24 (0.20–0.36) ng/mL *p* = 0.76; Figure 3). There was no difference between the severity groups in circulating CALR levels (*p* = 0.44, Figure 4). We observed a significant positive correlation between CALR levels and age (r = 0.35, *p* < 0.01) and BMI (r = 0.31, *p* < 0.01; Table 2). However, using multivariate regression analysis, none of the clinical variables were associated with altered CALR levels (all *p* > 0.05).

## 4. Discussion

To our knowledge this is the first study evaluating sLRP-1 and calreticulin levels in OSA. We reported lower levels of plasma sLRP-1 in OSA which were related to the severity of nocturnal hypoxia. There was no difference in calreticulin.

Decreased sLRP-1 levels in OSA can be explained by multiple mechanisms. Probably the most important regulator of LRP-1 expression is the SREBP-1 [10] which is upregulated by IH [3] and free fatty acids [23]. SREBP-1 is an important transcription factor involved in lipid homeostasis by controlling the expression of enzymes which are required for fatty acid, triglyceride and cholesterol synthesis [24]. Silencing of SREBP-1 resulted in an enhanced LRP-1 expression in human vascular smooth cells and in macrophages [10,25]. In addition, LRP-1 expression is induced by vitamin D [26] as well as by the klotho protein [27] which are both decreased in OSA [18,28]. Hypoxia inducible factor-1 alpha (HIF-1α) activated by IH is highly expressed in OSA [29,30,31,32]. Recently HIF-1α have been suggested to mediate OSA-associated metabolic abnormalities [33] and contribute to increased LRP-1 levels [34,35]. Our results suggest that the former mechanisms leading to decreased LRP-1 levels are predominant to the latter one. Moreover, age has a negative impact on the expression of LRP-1 in the liver, resulting in delayed clearance of chylomicron remnants in mice [36]. In line with this we detected a significant inverse relationship between sLRP-1 levels and age.

Although there is a close correlation between membrane-bound and soluble LRP-1 levels [37], shedding of LRP-1 is facilitated by matrix metalloproteases [37], atherogenic lipoproteins [38], proinflammatory cytokines [9], as well as clusterin [39] resulting in higher sLRP-1 levels. We detected an inverse correlation between sLRP-1 levels and markers of dyslipidaemia suggesting that the suppression of LRP-1 levels induced by free fatty acids was stronger than the shedding effect of atherogenic lipids. It is also possible that LRP-1 and sLRP-1 levels in OSA depend on the magnitude of dyslipidaemia as Calvo et al. detected a direct association between circulating sLRP-1 levels and total cholesterol and LDL-C in subjects with severe hypercholesterinaemia (LDL-C > 190 mg/dL) [38]. The current study population was comprised of subjects with normal LDL-C and total cholesterol levels. Of note, to mitigate this effect, our analyses were adjusted for lipid levels.

It appears that IH, rather than sleep fragmentation, was related to low sLRP-1 levels, as we did not find any relationship between sLRP-1 or calreticulin levels and markers of sleep quality. Of note, there is a growing evidence that sleep fragmentation is also independently related to metabolic abnormalities in OSA with a higher risk for cardiovascular diseases [40]. Our results needed to be considered carefully, as only half of the subjects had polysomnography as a diagnostic test. To assess if the analysed biomarkers are related to sleep fragmentation further studies are warranted. The relationships between markers of overnight hypoxaemia and sLRP-1 levels were weak, suggesting that plasma sLRP-1 concentrations are determined by multiple mechanisms, as discussed above.

CALR is a marker of immunogenic cells death [16]. Although hypoxia induces its translocation to the cell surface and extracellular release [41], we did not find a significant relationship between CALR levels and markers of IH. A previous study by Hirano et al. demonstrated that obesity without diabetes is associated with lower CALR expression [42] suggesting a counterbalancing mechanism leading to unaltered CALR levels in OSA.

Lower LRP-1 expression in OSA could contribute to metabolic dysfunction as well as augmented inflammation and coagulation. Inactivation of hepatic LRP-1 gene in mice resulted in decreased cholesterol-rich remnant lipoproteins clearance with consequential accumulation of these lipoproteins in the circulation [12], implicating that dyslipidaemia could be not only the reason for but the consequence of low LRP-1 levels. Supporting this, a significant inverse relationship was found between LRP-1 levels and lipid profile. LRP-1 also stimulates the expression of insulin receptor and the translocation of GLUT2 transporter [43]. In line with this, we found an inverse correlation between sLRP-1 levels and TyG, a marker of insulin resistance suggestive of the potential contribution of the decreased LRP-1 concentration to OSA-associated insulin resistance.

It has been reported that LRP-1 as well sLRP-1 deficiency led to increased levels of TNF-α [14,44,45], IL-1β [45], IL-6 [14], matrix metalloproteases [46] and monocyte chemoattractant protein type-1 [44], increased expression of inducible nitric oxide synthase [14] and it is also associated with elevated macrophage numbers [44] with low prevalence of M2 cells [47] in atherosclerotic plaques. LRP-1 also regulates the catabolism complement factors, such as C1s, C1r or C3 [14,48], hence reduced levels could lead to complement activation previously reported in OSA [49]. These studies indicate that reduced sLRP-1 levels could contribute to increased systemic inflammation in OSA.

It is also described that LRP-1 regulates the coagulation system on several points resulting in the attenuation of the coagulant activity. LRP-1 interacts with factor VIII (FVIII) which is the cofactor of factor Xa (FXa) initiating the intrinsic pathway. LRP-1 mediates the endocytosis of FVIII resulting its clearance from the circulation [50]. Moreover, factor FVIIIa-FXa complex and directly FXa may be inhibited by LRP-1 [51]. The levels of FVIII were measured in elevated concentration in patients with OSA [52]. Von Willebrand factor (vWF) is a transporter of circulating FVIII [53] and its elevated levels may play role in the increased platelet activation and aggregation in OSA [54]. LRP-1 is also involved in the clearance of vWF [55]. Furthermore, LRP-1 mediates the endocytosis of uPA [56] and uPAR [57], which are activators of the fibrinolysis. Taken together, decreased levels of LRP-1 may have an association with the OSA-related hypercoagulability.

The study has limitations. Obesity is a risk factor for the development of OSA. The prevalence of OSA is approximately 40% among overweight patients [58] and 70% of patients with OSA are obese [59]. Obesity itself leads to augmented inflammation and reduced antioxidant capacity [60]. During the patient recruitment we did not limit the BMI values and we included patients with various degrees of obesity. Thus, to reduce the effect of obesity, our primary analyses were adjusted for BMI. However, to eliminate this effect, studies involving non-obese patients with OSA are warranted. The study had a case-control design. To better understand the association between sLRP-1 and OSA, prospective trials, involving CPAP treatment are warranted. We believe that our findings could facilitate the planning of such trials. In this study PSG has been performed only in 39 cases, therefore our data on the markers of sleep architecture has to be interpreted carefully. Moreover, PSG is a more sensitive and specific tool to score arousals resulted from hypopnoea. Although fasting blood samples were taken for lipid measurements, diet, alcohol consumption or regular exercise were not controlled in this study and therefore it could be considered as a potential bias. These limitations could have contributed to small differences between control and OSA groups and weak correlations between sLRP-1 levels and markers of overnight hypoxaemia.

## 5. Conclusions

In summary, we reported lower sLRP-1 levels in patients with OSA independently of the gender, age, BMI or lipid profile. Thus, the dysregulation of the LRP-1 pathway could contribute to the metabolic and inflammatory changes as well as to hypercoagulability seen in OSA.

## Figures and Tables

**Figure 1 jcm-10-01494-f001:**
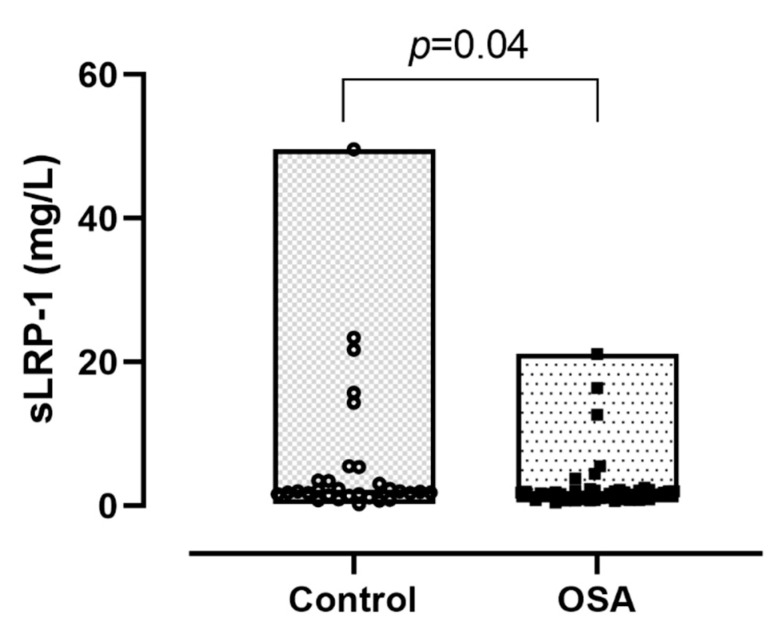
Soluble low-density lipoprotein receptor-related protein-1 (sLRP-1) concentrations between the control and obstructive sleep apnoea (OSA) group. Data are presented with median, minimum and maximum values.

**Figure 2 jcm-10-01494-f002:**
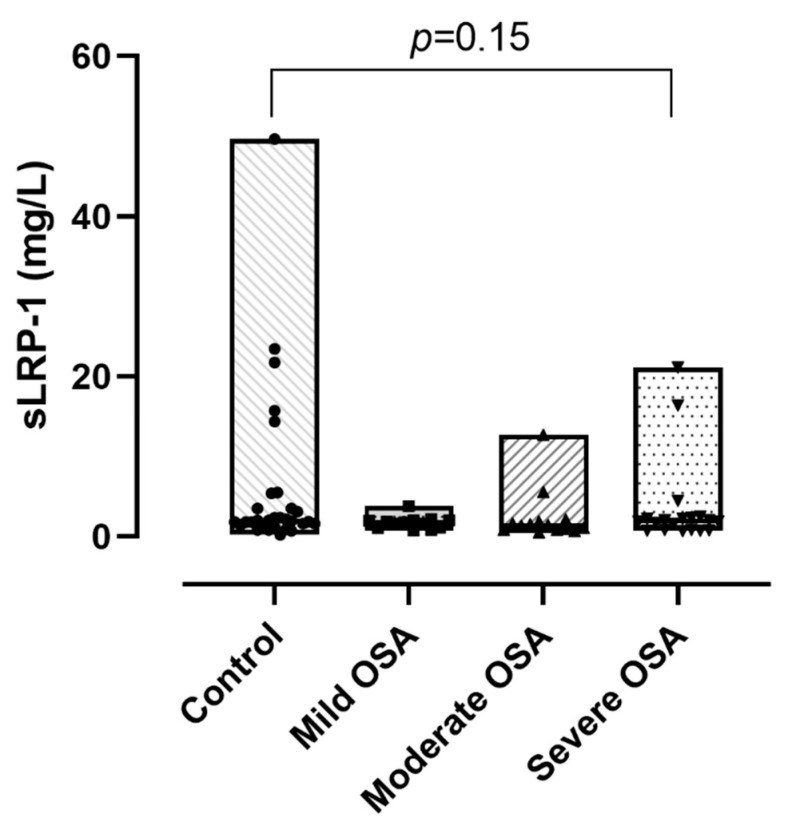
Soluble low-density lipoprotein receptor-related protein-1 (sLRP-1) concentrations between the severity groups of obstructive sleep apnoea (OSA). Data are presented with median, minimum and maximum values.

**Figure 3 jcm-10-01494-f003:**
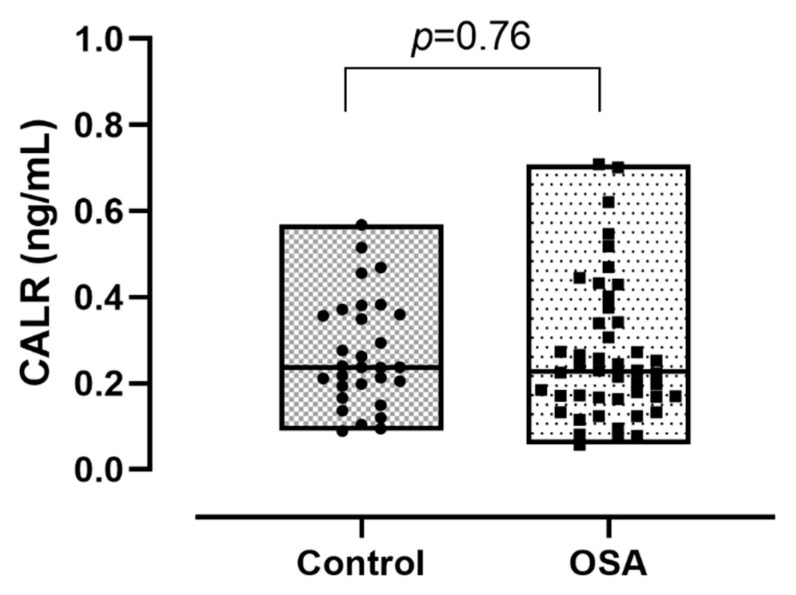
Calreticulin (CALR) concentrations between the control and obstructive sleep apnoea (OSA) group. Data are presented with median, minimum and maximum values.

**Figure 4 jcm-10-01494-f004:**
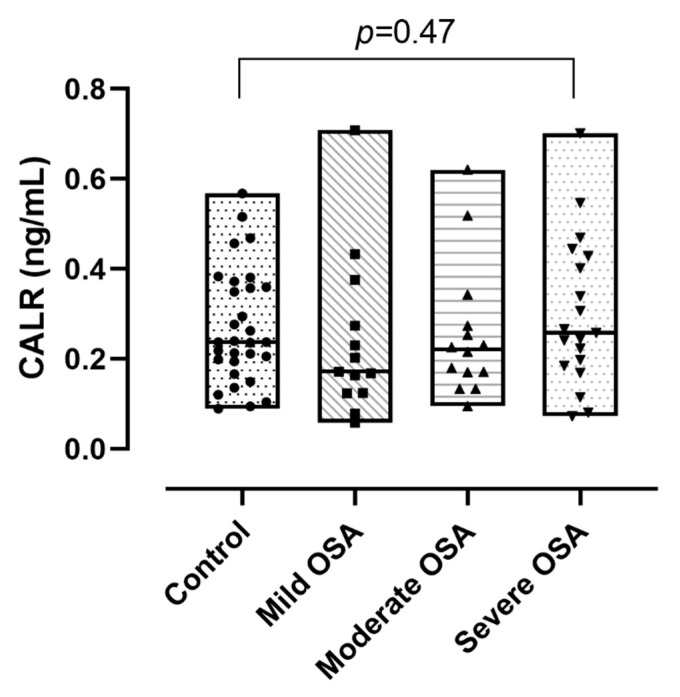
Calreticulin (CALR) concentrations between the severity groups of obstructive sleep apnoea (OSA). Data are presented with median, minimum and maximum values.

**Table 1 jcm-10-01494-t001:** Patients’ characteristics.

	Control (*n* = 30)	OSA (*n* = 46)	*p*
Age (years)	43 (30–51)	54 (46–62)	<0.001
Males (*n*, %)	23	67	<0.001
BMI (kg/m^2^)	23.89 (20.96–26.99)	29.89 (25.11–37.56)	<0.001
Hypertension (%)	30	67	0.001
Diabetes (%)	10	15	0.511
Dyslipidaemia (%)	27	30	0.723
Cardiovascular disease (%)	7	9	0.748
Cardiac arrythmia (%)	13	26	0.183
Smokers (%)	3	30	0.004
SBP (mmHg)	120.0 (110.0–128.8)	135.5 (122.0–148.0)	<0.001
DBP (mmHg)	70.0 (66.3–80.0)	84.0 (78.5–90.0)	<0.001
CRP (mg/l)	1.75 (0.84–2.50)	3.20 (1.79–4.79)	0.005
Glucose (mmol/l)	4.7 (4.3–5.2)	5.1 (4.9–6.2)	0.002
Cholesterol (mmol/l)	5.63 ± 1.03	5.59 ± 1.03	0.877
HDL-C (mmol/l)	1.59 (1.38–1.97)	1.25 (1.06–1.47)	<0.001
LDL-C (mmol/l)	3.39 ± 0.83	3.58 ± 0.88	0.328
Triglyceride (mmol/l)	1.29 (0.93–1.56)	1.64 (1.26–2.15)	0.004
Lipoprotein (a) (mmol/l)	0.22 (0.08–0.58)	0.35 (0.02–0.54)	0.937
ApoA1 (g/l)	1.66 (1.50–1.73)	1.46 (1.29–1.64)	0.097
ApoB (g/l)	1.17 (1.09–1.38)	1.19 (0.99–1.39)	0.790
TyG	8.37 ± 0.32	8.92 ± 0.50	<0.001
AIP	−0.13 ± 0.25	0.11 ± 0.25	<0.001
AHI (1/h)	2.15 (1.13–3.08)	26.05 (12.50–35.63)	<0.001
ODI (1/h)	0.90 (0.23–1.65)	22.0 (9.33–33.40)	<0.001
SPT (min)	431.56 ± 47.41	447.59 ± 41.43	0.270
TST (min)	401.31 ± 41.52	415.04 ± 31.75	0.249
TST90% (%)	0 (0–0)	4.5 (0.7–16.4)	<0.001
MinSatO_2_ (%)	91 (89–92)	83 (75–87)	<0.001
ESS	6 (4–8)	7 (5–10)	0.502

Data are presented as mean ± standard deviation or median (25–75% percentile). Apnoea-hypopnoea index (AHI), atherogenic index of plasma (AIP), apolipoprotein A1 (ApoA1), apolipoprotein B (ApoB), body mass index (BMI), C-reactive protein (CRP), diastolic blood pressure (DBP), Epworth Sleepiness Scale (ESS), high-density lipoprotein cholesterol (HDL-C), low-density lipoprotein cholesterol (LDL-C), minimal oxygen saturation (MinSatO_2_), obstructive sleep apnoea (OSA), oxygen desaturation index (ODI), systolic blood pressure (SBP), sleep period time (SPT), total sleep time (TST), total sleep time spent with oxygen saturation below 90% (TST90%), triglyceride glucose index (TyG).

**Table 2 jcm-10-01494-t002:** Results of the Spearman correlation.

	sLRP-1		CALR	
Age (years)	r = −0.292	*p* = 0.011	r = 0.345	*p* = 0.002
Male (*n*, %)	r = −0.061	*p* = 0.603	r = 0.023	*p* = 0.845
BMI (kg/m^2^)	r = −0.354	*p* = 0.002	r = 0.305	*p* = 0.007
Cigarette pack years	r = −0.306	*p* = 0.008	r = −0.039	*p* = 0.738
Glucose (mmol/L)	r = −0.143	*p* = 0.269	r = −0.077	*p* = 0.554
CRP (mg/L)	r = −0.212	*p* = 0.097	r = 0.207	*p* = 0.106
Cholesterol (mmol/L)	r = −0.174	*p* = 0.132	r = −0.027	*p* = 0.820
HDL-C (mmol/L)	r = 0.174	*p* = 0.134	r = −0.078	*p* = 0.503
LDL-C (mmol/L)	r = −0.233	*p* = 0.043	r = −0.030	*p* = 0.799
Triglyceride (mmol/L)	r = −0.265	*p* = 0.021	r = 0.139	*p* = 0.233
Lipoprotein (a) (mmol/l)	r = 0.105	*p* = 0.486	r = −0.234	*p* = 0.118
ApoA1 (g/L)	r = −0.062	*p* = 0.664	r = 0.029	*p* = 0.839
ApoB (g/L)	r = −0.211	*p* = 0.134	r = −0.024	*p* = 0.867
TyG	r = −0.367	*p* = 0.004	r = −0.008	*p* = 0.949
AIP	r = −0.269	*p* = 0.019	r = 0.125	*p* = 0.281
AHI	r = −0.177	*p* = 0.126	r = 0.041	*p* = 0.722
ODI	r = −0.233	*p* = 0.043	r = 0.121	*p* = 0.296
SPT (min)	r = −0.143	*p* = 0.385	r = −0.084	*p* = 0.611
TST (min)	r = −0.028	*p* = 0.865	r = 0.063	*p* = 0.702
TST90%	r = −0.194	*p* = 0.097	r = 0.116	*p* = 0.325
Min SatO_2_	r = 0.198	*p* = 0.091	r = −0.154	*p* = 0.190
ESS	r = 0.092	*p* = 0.440	r = −0.029	*p* = 0.811

Apnoea-hypopnoea index (AHI), atherogenic index of plasma (AIP), apolipoprotein A1 (ApoA1), apolipoprotein B (apoB), body mass index (BMI), calreticulin (CALR) C-reactive protein (CRP), Epworth Sleepiness Scale (ESS), high-density lipoprotein cholesterol (HDL-C), low-density lipoprotein cholesterol (LDL-C), minimal oxygen saturation (MinSatO_2_), oxygen desaturation index (ODI), soluble low-density lipoprotein receptor-related protein-1 (sLRP-1), sleep period time (SPT), total sleep time (TST), total sleep time spent with oxygen saturation below 90% (TST90%), triglyceride glucose index (TyG).

## Data Availability

The data are available from the corresponding author on request.

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
