# Peer review of "The Role of Soluble Low-Density Lipoprotein Receptor-Related Protein-1 in Obstructive Sleep Apnoea"

_jcm, 2021, doi:10.3390/jcm10071494_

Round 1

Reviewer 1 Report

This is a very interesting and well-done study of participants 76 participants (46 with OSA and 30 controls) looking at the correlation with the presence of OSA with regards to inflammatory markers, in this case sLRP-1.  What was found includes that sLRP-1 is associated with the presence of OSA, which is in itself associated with several metabolic and inflammatory abnormalities.  While it is very taboo to do so, I do not have much to comment on here.  This was well-written and concise, and any limitations I would have thought of (not all participants undergoing PSG), the authors addressed. 

Author Response

The authors would like to thank for this comment of the Reviewer. 

Reviewer 2 Report

The manuscript entitled „The role of soluble low-density lipoprotein receptor-related protein – 1 in obstructive sleep apnoea” reports on concentration of sLRP-1and CARL in OSA patients compared to healthy controls and names independent independent predictive parameters for lowered sLRP-1in OSA.

DescribingsLRP-1 and CARL as biomarkers seems a little inadequate.

The inclusion/exclusion criteria shou include age, BMI information and time patient had to spent in sleep during PSG examination.

Instead of slashes please use brackets.

The discussion should be expended towards IH as a key factor in the upregulation of sLRP-1(doi: 10.1161/ATVBAHA.111.225490). Especially that in recent years several papers have shown increase in HIF-1alpha in OSA patients (doi: 10.5664/jcsm.8682, doi: 10.20452/pamw.15104, doi: 10.17305/bjbms.2016.1579, doi: 10.3390/jcm9051599) and its possible influence on glucose metabolism (doi: 10.3389/fphys.2020.01035).

Author Response

Q1: The inclusion/exclusion criteria should include age, BMI information and time patient had to spent in sleep during PSG examination.

A1: Thank you for your suggestions. We included the requested information about the age and the minimal time spent in sleep during the sleep study (Page 2, Paragraph 5). The BMI values were not taken into consideration as inclusion- or exclusion criteria. Thus, we mentioned this as a limitation in the Discussion (Page 9, Paragraph 6).

Q2: Instead of slashes please use brackets.

A2: We replaced the slashes to brackets in the Abstract and in the Results section where they were used previously (mainly in Table 1). 

Q3: The discussion should be expended towards IH as a key factor in the upregulation of sLRP-1 (doi: 10.1161/ATVBAHA.111.225490). Especially that in recent years several papers have shown increase in HIF-1alpha in OSA patients (doi: 10.5664/jcsm.8682, doi: 10.20452/pamw.15104, doi: 10.17305/bjbms.2016.1579, doi: 10.3390/jcm9051599) and its possible influence on glucose metabolism (doi: 10.3389/fphys.2020.01035).

A3: Thank you for raising this important point. We extended the Discussion with the requested information and added the requested references (Page 8, Paragraph 2).

Reviewer 3 Report

In this work authors evaluated the levels of sLRP-1 and calreticulin in a group of patients with OSA and compared them with control group. They observed that sLRP-1 was reduced in OSA individuals, but not differences were found in calreticulin between the same groups. Study is well structured, well written and the design is correct. However I have some concerns to comment:

  1. Table 1 (column “Total”). I believe that this column can be confusing to data interpretation, so I suggest deleting it. Also, it also would be interesting to perform a table with the same parameters comparing the different severity groups (mild, moderate and severe OSA). This table could be placed in supplementary materials.
  2. Page 5 (Lines 177-184): this paragraph must be placed as footnote of Table 1.
  3. In subsections 3.2 and 3.4 (Figures 1 and 2) the change of graphs could help to visualize the data, showing all points and the minimum to maximum in a box and whiskers plot. Also, a figure of sLRP-1 and calreticulin in the different OSA severities could be interesting to include, although not significant differences were observed..
  4. In order to help to reader’s understanding, correlations value should be presented in a Table.
  5. According to correlations values, although you observed significant differences, the correlation coefficients are mild-moderate. It should be taken into account and discussed, in order to avoid risky conclusions.

Author Response

Q1: Table 1 (column “Total”). I believe that this column can be confusing to data interpretation, so I suggest deleting it. Also, it also would be interesting to perform a table with the same parameters comparing the different severity groups (mild, moderate and severe OSA). This table could be placed in supplementary materials.

A1: Thank you for your valuable suggestions. We deleted column “Total” in Table 1 as requested. Patient characteristics and comparisons of the OSA severity groups have been recently added, as Table S1 in the Supplementary material.

Q2: Page 5 (Lines 177-184): this paragraph must be placed as footnote of Table 1.

A2: Thank you for your note. It was originally the footnote of Table 1. We reformatted this paragraph to avoid further misunderstanding.

Q3: In subsections 3.2 and 3.4 (Figures 1 and 2) the change of graphs could help to visualize the data, showing all points and the minimum to maximum in a box and whiskers plot. Also, a figure of sLRP-1 and calreticulin in the different OSA severities could be interesting to include, although not significant differences were observed.

A3: Thank you. The Figures have been modified as requested and new figures have been created where biomarker levels are plotted along severity subgroups.

Q4: In order to help to reader’s understanding, correlations value should be presented in a Table. According to correlations values, although you observed significant differences, the correlation coefficients are mild-moderate. It should be taken into account and discussed, in order to avoid risky conclusions.

A4: Thank you for your suggestion. We added Table 2 with the correlation values (Page 6). We expanded the discussion with these limitations (Page 9, Paragraph 1).

Round 2

Reviewer 3 Report

Authors have solved succesfully all my previous questions.